# Immunological Effects of a Single Hemodialysis Treatment

**DOI:** 10.3390/medicina56020071

**Published:** 2020-02-12

**Authors:** Andrea Angeletti, Fulvia Zappulo, Chiara Donadei, Maria Cappuccilli, Giulia Di Certo, Diletta Conte, Giorgia Comai, Gabriele Donati, Gaetano La Manna

**Affiliations:** Department of Experimental Diagnostic and Specialty Medicine (DIMES), Nephrology, Dialysis and Renal Transplant Unit, S. Orsola-Malpighi Hospital, University of Bologna, 40138 Bologna, Italy; andrea.angeletti6@unibo.it (A.A.); fulvia.zappulo@studio.unibo.it (F.Z.); chiara.donadei@studio.unibo.it (C.D.); maria.cappuccilli@unibo.it (M.C.); giulia.dicerto@studio.unibo.it (G.D.C.); diletta.conte2@unibo.it (D.C.); giorgia.comai@aosp.bo.it (G.C.); gabriele.donati@aosp.bo.it (G.D.)

**Keywords:** complement, hemodialysis, immune response, inflammation, kidney transplant, lymphocytes, single dialysis session

## Abstract

Immune disorders, involving both innate and adaptive response, are common in patients with end-stage renal disease under chronic hemodialysis. Endogenous and exogenous factors, such as uremic toxins and the extracorporeal treatment itself, alter the immune balance, leading to chronic inflammation and higher risk of cardiovascular events. Several studies have previously described the immune effects of chronic hemodialysis and the possibility to modulate inflammation through more biocompatible dialyzers and innovative techniques. On the other hand, very limited data are available on the possible immunological effects of a single hemodialysis treatment. In spite of the lacking information about the immunological reactivity related to a single session, there is evidence to indicate that mediators of innate and adaptive response, above all complement cascade and T cells, are implicated in immune system modulation during hemodialysis treatment. Expanding our understanding of these modulations represents a necessary basis to develop pro-tolerogenic strategies in specific conditions, like hemodialysis in septic patients or the last session prior to kidney transplant in candidates for receiving a graft.

## 1. Introduction

Patients with end stage renal disease (ESRD) are known to have an increased risk of cardiovascular morbidity and mortality, related to traditional (diabetes mellitus, hypertension, dyslipidemia, and old age) and non-traditional risk factors, including systemic inflammation and immune deficiency [1,2,3,4].

Patients affected by ESRD receiving chronic hemodialysis usually present immune disorders, not solely related to the primary renal disease, which involve both innate and adaptive immune systems [5]. The consequent imbalance between pro- and anti-inflammatory mechanisms enhances the risk for cardiovascular disease, infections, and malignancy [6,7], accounting for the elevated morbidity and mortality rates in uremic patients [8,9].

Kidney function loss is progressively accompanied by a reduction of naïve T cells, resulting in an inadequate immune response to antigens [10]. Previous studies have proven that patients under chronic hemodialysis treatment have immune dysfunction compared to healthy subjects, mainly due to the pro-inflammatory phenotypes of monocytes [11], the inverted CD4+/CD8+ ratio, the accumulation of differentiated T cells, and a significant reduction of naïve T cells [12,13,14]. These immune disturbances are responsible for several effects, namely increased susceptibility to developing bloodstream infections [15] or deficiencies in the immune response to vaccinations [16].

However, while there are plenty of data on the immunological effects of chronic hemodialysis, how the single hemodialysis treatment acts on the immune system is still far to be completely elucidated. A transient immune effect of hemodialysis may have a potential impact in different settings, such as delayed graft function in the early stages of kidney transplant or acute kidney injury during sepsis. This narrative review summarizes the current knowledge on the immunogenicity of the single hemodialysis treatment, with a particular focus on innate and adaptive immunity.

## 2. Innate Immune Response During a Single Hemodialysis Treatment

### 2.1. Complement System

The complement system is part of the innate immune response with the main function of protecting the host from foreign pathogens [17,18,19]. The complement cascade is activated by the lectin pathway, the classical pathway and the alternative pathway, all of them converging on an enzymatic multimeric protein complex, the C3 convertase [18]. C3 cleavage produces C3a and C3b, the latter eliciting the formation of C5 convertase. C5 cleavage leads then to the assembly of the membrane attack complex (MAC, C5b-9). Besides MAC, soluble and surface-bound split products, including C3a, C3b, iC3b, C3dg, and C5a, are implicated in inflammatory response [20].

In 1977, Craddock et al. firstly described the possible pathological mechanisms for acute pulmonary dysfunction, a cardiopulmonary complication occurring in the first phase of the dialysis session with cellophane-membrane, as a consequence of a complement-mediated leukostasis (Table 1) [21]. The authors observed a severe transient neutropenia and a reduction of monocytes in the first hour of treatment, with activation of C3 and factor B, another serum protein of the alternative pathway [18]. During hemodialysis sessions, the dialysis membrane or the catheters may be coated with complement components, followed by neutrophil adhesion [22]. Several studies agree that in the course of hemodialysis sessions, the C3/C3d ratio and soluble C5b-9 in plasma increase up to 70% [23,24,25]; however, which complement pathway is mainly involved in the activation of complement cascade is still to be defined.

The alternative pathway is known to affect immunity during hemodialysis. In healthy conditions, the alternative pathway is continuously activated by the hydrolysis of C3 on cell surfaces and this activation is balanced by circulating and cell surface regulatory factors [20]. During hemodialysis treatment, a transient loss of complement inhibitors via absorption from dialysis membrane can generate a temporary break of the stability between positive and negative regulators of the complement cascade. A proteomic analysis by Mares et al. showed that hemodialysis polysulfone membranes adsorb complement inhibitor factor H and clusterin (Table 1) [26]. Factor H is an inhibitor of C3 convertase, while clusterin prevents terminal cascade activation blocking the formation of C5b-9. The recruitment of these inhibitors by the dialysis membrane might therefore trigger complement activation. The absorption of properdin to the dialyzer, the only known positive regulator of complement activation that stabilizes the C3bBb complex leading to increased activity of the alternative pathway [18], corroborates the hypothesis on the contribution of alternative pathway activation during hemodialysis [27].

Likewise, the lectin pathway is involved in complement activation during hemodialysis procedure. Filcolin-2, one of the few molecules known to be a trigger of the lectin pathway [28], was measured through mass spectrometry assay in the plasma and in the eluates of 16 patients during hemodialysis with polysulfone membranes: the ficolin-2 eluate/plasma ratio significantly increased during the treatment, suggesting a possible specific adsorption by the membrane [26]. Similar findings were reported in a different tandem mass spectrometry on eluates and serum samples of five patients during the hemodialysis session: the eluate intensity of ficolin-2, clusterin, complement C3c fragment, and apolipoprotein A1 differed significantly from the counterpart serum levels, implying preferential adsorption to the polysulfone dialyzers [27]. The reduction of such factors in bloodstream was associated with a serum increase of C5a and leukopenia during hemodialysis, and a higher incidence of cardiovascular events was found in patients with lower serum levels of mannose binding leptin (MBL) [29,30]. Notably, this relationship was not correlated with other traditional cardiovascular risk factors (Table 1) [30].

Lastly, the classical pathway may also contribute to the complement activation through the adhesion of C1q on membrane-adsorbed immunoglobulin G (IgG), although conflicting results are available in the literature [31].

The activation of complement cascade may be further sustained by the covalent binding of C3b on nucleophilic surface, characterized by the combination of oxidized lipids of blood vessels and dialysis synthetic or cellulosic polymeric membranes coated with albumin, IgG, lipopolysaccharide, and other non-pathological bacteria commonly present in water or dialysis solutions [32].

In conclusion, despite the remarkable developments in the biocompatibility of hemodialysis, complement activation remains a significant concern in the single hemodialysis treatment, mainly through alternative pathway and lectin pathway. Short-term effects of the complement system are the enhancement of inflammation and coagulation, while long-term complications include fibrosis and cardiovascular events.

### 2.2. Cellular Components

Besides the lack of a full comprehension of the immune reactivity against biomaterials, emerging evidence indicates that innate immune cells, mostly neutrophils and macrophages, are essential players in the initial phases of the immune response [33].

While blood flows through the dialysis membranes, synthetic materials elicit a foreign body reaction characterized by recruitment of neutrophils and monocytes [34,35], which release pro-inflammatory cytokines, namely interleukin 1 (IL-1), interleukin 6 (IL-6), and tumor necrosis factor-α (TNF-α) [36,37]. Therefore, the measurement of activated monocytes provides useful information on dialyzer biocompatibility [38].

Previous in vitro studies explored the effect of inflammatory molecules released by activated monocytes on endothelial cells (ECs), co-culturing CD14+ CD16+ monocytes with human umbilical vein endothelial cells (HuVECs). Increased apoptosis and radical-oxygen-species (ROS) activity was reported in HuVECs co-cultured with monocytes [39], confirming the hypothesis that oxidative stress triggered by microinflammation in the ECs results in endothelial injury [40]. Successive evidence in humans revealed how CD14+CD16+ monocytes derived from chronic kidney disease patients have an increased adhesion ability to ECs [41]. Moreover, hemodialysis patients display higher relative counts of CD14+CD16+ monocytes compared to peritoneal dialysis patients, and a relationship was found between increased CD14+CD16+ monocytes and endothelial damage [42]. Clinical studies investigating different types of hemodialysis membranes report indeed discordant results on the efficacy of the different dialyzers in reducing markers of inflammatory and procoagulant state, namely molecules involved in leukocyte, platelet and endothelial activation [43,44]. A single hemodialysis session with low-flux membranes (cellulose-based and polysulfone membranes) did not seem to affect blood levels of TNF-α, interleukin-6, P-selectin, and coagulation factors [43]. In an observational prospective study on 30 patients undergoing hemodialysis with high-flux polymers, polysulfone or polyethersulfone, Martinez-Miguel et al. evaluated whether the use of more biocompatible membranes could reduce monocyte activation and the in vitro ability of circulating cells to damage endothelial monolayers. When compared to two weeks of hemodialysis with low-flux polysulfone, the use of polyethersulfone membrane resulted in decreased adhesion of monocytes to ECs, reduced monocyte-induced cellular toxicity and lower expression of endothelin-converting enzyme-1 (ECE-1), a protein involved in endothelin synthesis. However, these effects were not reported after a single hemodialysis treatment, suggesting a cumulative effect that might be explained by the continuous use of such high-quality dialyzers (Table 1) [45].

Besides dialysis membranes, the synthetic vascular access also contributes to the modulation of inflammatory response during the hemodialysis session. In a prospective observational study enrolling 100 chronic hemodialysis patients, we previously demonstrated that such condition is characterized by an increased inflammatory state: compared to healthy subjects, the hemodialysis patients revealed lower serum albumin levels, increased C-reactive protein (CRP), lower absolute monocyte count, but increased expression on monocyte surface of cell activation marker CD14. When comparing the patients grouped according to the type of vascular access, we found that the ones with arteriovenous graft and tunneled-cuffed-catheter displayed higher circulating levels of inflammation markers (CRP, IL-6, and TNF-α), increased expression on monocyte surface of cell activation markers (CD14, CD44, and CD32), and higher relative number of monocytes with a senescent phenotype (CD14+ CD32+) than those with a native fistula [46]. These results probably reflect the prolonged and continued use of central venous catheter or graft, which determines either the contact of the monocytes with an artificial material or monocyte activation after subclinical infection [47,48].

Taken together, all these data suggest that the type of vascular access may play a differential role in promoting an inflammatory response. However, despite the presence of some limited experiences, exhaustive reports describing the defined effects of a single hemodialysis session on white blood cells are still lacking.

Finally, also the quality of dialysis water represents one of the modifiable uremia-induced risk that could cause inflammation [49]. The current systems for water treatment are not able to remove endotoxins, bacteria and their products or short bacterial DNA fragments, that given their small size, easily pass thought the dialyzer membrane into the bloodstream [50].

We have previously reported that hemodialysis patients treated with ultrapure dialysate had lower rate of lesions from β2-microglobulin-related amyloidosis and reduced CRP compared to those who had undergone dialysis with no ultrapure dialysate. These findings can be explained by the better bacteriological quality of the water [51].

In order to investigate the possible pro-inflammatory effect of dialysis water, Di Iorio et al. investigated the ability of an ultrafilter to improve the quality of dialysis water and the effect on cytokine profiles in 33 stable hemodialysis patients. The authors observed decreased levels of pro-inflammatory cytokines and increased levels of anti-inflammatory cytokines in patients treated with ultrapure dialysate [52].

Moreover, Kwan et al. demonstrated that ultrapure dialysate resulted in lower levels of circulating endotoxins with no differences in bacterial DNA levels, which resulted in less severe vascular stiffness and improvement of systemic inflammation [53].

These results support the hypothesis that circulating bacteria contribute to maintaining a systemic inflammatory status.

## 3. Adaptive Immune Response during Hemodialysis Treatment

### 3.1. T Cells

Several studies have shown the alteration of T cell function in ESRD [54], but the mechanisms are still to be clarified, and may be only partially explained as the consequence of higher levels of proinflammatory cytokines in uremic serum. It is known that conventional T cells are reduced in ESRD patients due to the increased apoptosis rate and reduced proliferation [55]. However, pathogenetic hypotheses on T cells dysfunction are not definitive and need more experimental data.

Mansouri et al. described a marked reduction of Th2 and regulatory T cells (Tregs) in patients receiving chronic hemodialysis compared to healthy subjects [56].

More recently, Lisowska et al. investigated the effect of a single hemodialysis session on T lymphocyte subsets, and observed an increased CD4+/CD8+ ratio as a consequence of the reduction of CD8+ T cells and not of the increase of CD4+ T cells (Table 1) [57]. These findings are consistent with a previous study indicating that hemodialysis procedures are, at least in part, responsible for the development of T cell lymphopenia due to the induction of apoptosis (Table 1) [58]. Data from our group and other authors highlighted the immunological effect of erythropoietin (EPO) in counteracting T cell effects: EPO ligation of its receptor on CD4+ T cells has been proven to directly prevent Th17 generation and to induce trans-differentiation of Th17 into IL-17-Foxp3+ CD4+ T cells, downregulating the pro-inflammatory T cell response [59,60,61,62,63,64,65]. Recombinant human erythropoietin (rhEPO) is usually administered at the end of hemodialysis treatment, therefore it represents a further variant to consider in the modulation of the immune system of the single hemodialysis.

The reiteration of the above described phenomena during the many hemodialysis sessions that ESRD patients chronically receive may lead to the exhaustion of the lymphocyte activation capacity, a phenomenon known as stress-induced premature senescence [14]. Litjens et al. postulated that the drop in the number of naïve T cells in hemodialysis is explained by the impaired thymic output of naïve T cells with deficient proliferation and disrupted homeostasis in the periphery [66]. Moreover, the same group demonstrated that the decline of renal function is correlated with the activation of premature T cell ageing of both CD4+ and CD8+ T cells [67]. Hemodialysis is also associated with a moderate increase of memory T cell senescence in patients over 50 years of age [14]. A recent investigation by Almeida et al. in a cohort of 17 patients with diabetic kidney disease undergoing hemodialysis reported that a single hemodialysis treatment triggered activation of total T cells, in particular CD8+ CD25+ lymphocytes, a subset of activated effector T cells (Table 1) [68]. Unfortunately, the lack of characterization with regulatory T cell markers, such as Foxp3, does not allow us to draw further conclusions from this finding.

Hence, hemodialysis sessions represent a really important factor contributing to the immune deficiency usually described in patients with ESRD but, on the other hand, they also seem to stimulate T cell activation, perhaps secondary to the effects of innate immune factors, like complement components and monocytes.

### 3.2. B Cells

B lymphocytes are in charge of the adaptive immune response by secreting antibodies following contact with antigens. In the T cell-dependent response, antigens activating cells need the help of T cells, which can be impaired by hemodialysis treatment. Therefore, hemodialysis is generally associated with a diffuse reduction of B cell subpopulations [69].

In a case–control prospective observational study, Pahl et al. proved how patients receiving chronic hemodialysis treatment are characterized by lymphopenia, including a reduced count of B lymphocytes, when compared to healthy subjects, despite similar serum levels of B-cell activating factor (BAFF) and interleukin-17 (IL-17) between the two groups. The authors also reported a reduced expression of BAFF and IL-17 receptors on the surface of CD19+ CD10+ B-cells. Moreover, no differences in terms of apoptosis of B cells between hemodialysis patients and control subjects were found [70]. In contrast, in a case–control study including 36 hemodialysis patients and 32 pre-dialysis uremic subjects, an increased apoptosis rate of B cells was observed in hemodialysis group, probably due to the reduced expression of Bcl-2, a regulator protein that modulates cell death [71]. In a nutshell, the reduction of B cells may be a consequence of both the increased apoptosis and downregulation of BAFF receptors. In hemodialysis patients, despite their reduced number, B cells have a significant ability to release pro-inflammatory cytokines, like TNF-α and IL-6 [36]. A more recent study, analyzing the immune phenotype of B cells in 27 hemodialysis patients vs healthy subjects, reported a significant decline of immature B cells and an increase of memory B cells [72]. Nevertheless, the current knowledge of the effects of a single hemodialysis session on B cells is still limited.

## 4. Possible Role of Hemodialysis in Early Stages of Kidney Transplant: Pro- or Anti-Tolerogenic Effects?

Kidney transplant represents the best treatment option for ESRD. However, due to the scarcity of deceased and living donors, the majority of ESRD patients need to receive renal replacement therapy for a median time of 5 years, according to the United Network for Organ Sharing (UNOS) in the United States. It is well established that the time spent on the waiting list impacts negatively on long-term outcomes of transplant [73]. On the other hand, the impact of dialysis modality is still debated. It has been recently reported that pre-transplant patients on peritoneal dialysis show a significantly lower risk of delayed graft function after transplant than hemodialysis patients [74,75].

In a recent observational study on 15 patients eligible for renal transplantation receiving hemodialysis, Mai et al. investigated leukocytes subsets, complement activation, and concentrations of several pro-inflammatory cytokines before, during and after a hemodialysis session. In the first hour, they described a rise in CD4+ T cells and activated CD4+ HLA-DR+ T cells, with a decrease of CD8+ and CD8+ HLA-DR+ subsets. The counts of CD3+ lymphocytes expressing the early activation marker CD25 were also increased over the whole observation period. Moreover, the authors reported a reduction in naïve T cells during hemodialysis, feasibly due to the activation and subsequent differentiation to effector or memory cells after contact with the dialysis membrane. Furthermore, they observed a rise in CD8+ CCR5+ T cells that are crucial for the migration of lymphocytes. Tregs and HLA-DR+ Tregs increased significantly in the course of hemodialysis. In summary, the authors showed an increase in cells generally responsible for graft rejection and a slight rise in cells that induce tolerance. Based on previous results, Mai and colleagues suggest avoiding short-time dialysis before kidney transplantation (Table 1) [76]. Different studies have reported contrasting results on the effect of pre-transplant hemodialysis modality on kidney transplant outcomes [77], but further investigations are necessary to draw a definitive conclusion.

## 5. Single Hemodialysis Treatment and Immune System: Current Knowledge and Future Insights

Nowadays, the approach to dialysis in common clinical practice often follows a “one-size-fits-all” rule rather than a prescription of the best hemodialysis treatment tailored to different types of patients (Table 2). A better comprehension of the changes occurring in innate and adaptive immune responses during a single hemodialysis treatment (Figure 1) is essential for providing a personalized therapy, based on renal disease and comorbidities [78,79], also considering the synergistic contribution of uremia and the hemodialysis procedure itself to impaired immune function in ESRD.

While recent research and technology has led to great improvements in the field of dialysis membranes, less information is available on the immunological reactivity related to the single dialysis session, mostly derived from old studies with poorly representative populations, where patients were treated with less biocompatible dialyzers compared to the synthetic alternatives currently in use [80]. Increasing our awareness on this issue will open a new scenario for the development of pro-tolerogenic strategies in particular clinical settings, like hemodialysis in septic patients or the last dialysis session prior to kidney transplantation in candidates for receiving a graft.

## 6. Conclusions

To summarize, although the effect of the single session on immune response is still far to be fully elucidated, emerging evidence indicates the existence of an interplay between mediators of innate and adaptive response, above all complement cascade and T cells.

For these reasons, it is essential to focus future research on the characterization of the acute responses of the innate and the adaptive immune system in light of the novel biocompatible hemodialysis membranes.

## Figures and Tables

**Figure 1 medicina-56-00071-f001:**
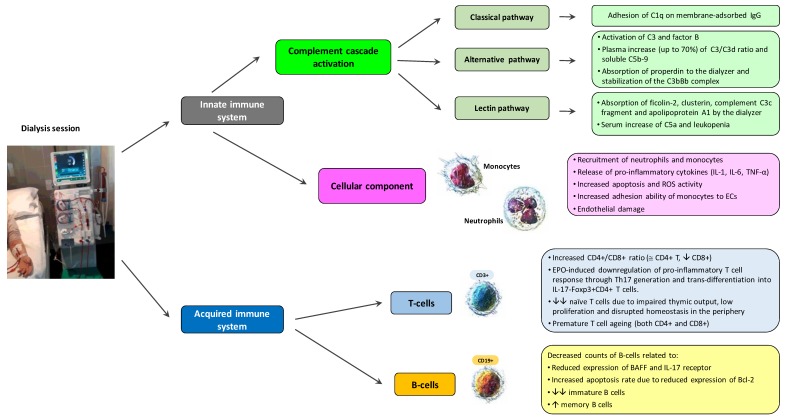
Effects of hemodialysis on innate and acquired immunity. BAFF, B-cell activating factor; EC, endothelial cells; EPO, erythropoietin; IL-1, interleukin 1; IL-6, interleukin 6; ROS, radical-oxygen-species; TNF-α, tumor necrosis factor alpha.

**Table 1 medicina-56-00071-t001:** Major published observational clinical studies reporting the effects of single hemodialysis treatment on the immune system.

Reference	Study Design	Patients	Endpoints	Results
**Innate Immunity**				
Craddock PR, 1977	Observational prospective	34 hemodialysis patients with leukopenia during hemodialysis sessions	Detection of mechanisms for acute pulmonary dysfunction reported in the first hour of hemodialysis with cellophane membranes	15/34 had impaired pulmonary function with transient neutropenia and reduction of monocytes during the first hour of hemodialysis sessions with activation of C3 and factor B
Mares J, 2010	Observational prospective	16 hemodialysis patients with polysulfone dialyzers	Leukocyte counts and complement components levels were monitored during hemodialysis in serum and equates	C3c, ficolin-2, mannan-binding lectin serine proteases, and properdin were enriched in equates and decreased in serum.
Poppelaars F, 2018	Observational prospective	55 hemodialysis patients	To correlate cardiovascular event with plasma levels of MBL, properdin, and C3d/C3 ratio	Lower levels of MBL and properdin in patient cardiovascular events
**Adaptive Immunity**				
Borges A, 2011	Cross-sectional	47 hemodialysis patients (12 evaluated before and after hemodialysis sessions)	Characterization of T lymphocyte phenotype and apoptosis	The hemodialysis procedure contributed to the development of T-cell lymphopenia, at least in part, by apoptosis induction of CD8+ T cells
Martinez-Miguel P, 2014	Observational prospective	30 hemodialysis patients with high-flux polymers	To evaluate monocyte activation	No change after a single hemodialysis treatment
Almeida A, 2015	Observational prospective	17 hemodialysis and diabetic patients	To assess the expression of T cell activation markers and quantify inflammatory cytokines before and after a single hemodialysis session	CD25+ cells and CD8+ CD25+ increased significantly, while CD69 T cells and CD4+ CD25+ significantly decreased after the hemodialysis session
Lisowska KA, 2019	Observational prospective	14 hemodialysis patients	To investigate the effect of the single hemodialysis session on T lymphocyte subsets	Increased CD4+/CD8+ ratio as a consequence of the reduction of CD8+ T
Mai K, 2019	Observational prospective	15 ESRD patients receiving hemodialysis treatment before renal transplant	Alterations of adaptive immune response to polynephron membrane	During the first hour there was an increase in CD4+ T cells and activated CD4+ HLA-DR+ T cells, with a decrease of CD8+ T cells and CD8+ HLA-DR+ T cells

ESRD, end stage renal disease; MBL, mannose binding leptin.

**Table 2 medicina-56-00071-t002:** Current criteria used in our Center for the prescription of hemodialysis treatment.

	Normal	Diabetes Albumin < 4 gr%	Diabetes Albumin < 4 gr% Cardiac Disease Hypotension	Allergies
HD Technique	HD low/high flux	HD high flux	On-line HDF (convection target > 20 L/session)	High Flux HD On-line HDF (convection target > 20 L/session)
Dialyzer	Synthetic	Synthetic	Synthetic	Bisfenol-free PVP-free
Dialysate flow	500 mL/min	500 mL/min	500 mL/min	500 mL/min
Dialysate composition	Na^+^ 140 mEq/L K^+^ 2.5–3 mEq/L Ca^++^ 1.5 mmol/L	Na^+^ 140 mEq/L K^+^ 2.5–3 mEq/L Ca^++^ 1.5 mmol/L	Na^+^ 140 mEq/L K^+^ 2.5–3 mEq/L Ca^++^ 1.5 mmol/L	Na^+^ 140 mEq/L K^+^ 2.5–3 mEq/L Ca^++^ 1.5 mmol/L
Ultrafiltration rate	max 10 mL/kg/hour	max 10 mL/kg/hour	max 10 mL/kg/hour	max 10 mL/kg/hour
Anticoagulation	LMWH	LMWH	LMWH	LMWH
Length (according to clinical status)	210-270 min	210-270 min	210-270 min	210-270 min
Rhythm (according to clinical status)	1->2->3->4 times/week	1->2->3->4 times/week	1->2->3->4 times/week	1->2->3->4 times/week

HD, hemodialysis; HDF, hemodiafiltration; LMWH, low-molecular-weight heparin; PVP, polyvinylpyrrolidone.

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
