# Peer review of "Immunological Effects of a Single Hemodialysis Treatment"

_medicina, 2020, doi:10.3390/medicina56020071_

Round 1

Reviewer 1 Report

This extensive and well written review on the immunological effects of dialysis sessions and potential deleterious consequences should also include to fully cover the topic  the well-demonstrated pro-inflammatory effect of non ultra pure dialysate (versus ultra pure dialysate) and the emerging potential beneficial effect on inflammatory and immune parameters of HDx with MCO membrane.    

Author Response

Dear Reviewer,

The manuscript has been modified in consideration of your comments, and our replies are below each point. We have also updated the bibliography accordingly. The number of pages and lines refers to new version of the manuscript.

On behalf of myself and my coauthors, I would like to thank you for the suggestions which allowed us to make the paper more comprehensible and hopefully to improve it. 

Yours sincerely,

Prof. Gaetano La Manna

This extensive and well written review on the immunological effects of dialysis sessions and potential deleterious consequences should also include to fully cover the topic  the well-demonstrated pro-inflammatory effect of non ultra pure dialysate (versus ultra pure dialysate) and the emerging potential beneficial effect on inflammatory and immune parameters of HDx with MCO membrane.  

This is an interesting point, so we have added a specific paragraph at the end of 2.2. section (page 4, lines 160-177).

Reviewer 2 Report

The review is interesting and well-written. I have some comments for the authors to improve the global interest of reading such a review:

The review is really good but you need an outbreaking point of view about what you're describing inside.

I suggest to had Sum up figures for effects on innate an acquired immunity.

I suggest to had either a paragraph for the unmet needs about this activation or a sum up paragraph about what we don't know and what should be upgraded. I'm totally agree with the conclusion (one-fit-for-all position) and that clearly need to be highlighted. Personalized prescription need to be the standard and effects on immune system need to be included.

You could maybe suggest a work up and a decisional algorithm for dialysis prescription, that could be great.

Minor comment :

Table 1: put it on a landscape special page to upgrade the readability

Author Response

Dear Reviewer,

The manuscript has been modified in consideration of your comments, and our replies are below each point. We have also updated the bibliography accordingly. The number of pages and lines refers to new version of the manuscript.

On behalf of myself and my coauthors, I would like to thank you for the suggestions which allowed us to make the paper more comprehensible and hopefully to improve it. 

Yours sincerely,

Prof. Gaetano La Manna

The review is interesting and well-written. I have some comments for the authors to improve the global interest of reading such a review:

The review is really good but you need an outbreaking point of view about what you're describing inside.

I suggest to had Sum up figures for effects on innate an acquired immunity.

We added Figure 1 at page 9, representing schematically and synthetically the main changes on the innate and adaptive immunity occurring in course of a dialysis session.

I suggest to had either a paragraph for the unmet needs about this activation or a sum up paragraph about what we don't know and what should be upgraded.

We addressed this interesting point, adding some general considerations and conclusive remarks in a specific section before the final conclusions (chapter 5: “Single hemodialysis treatment and immune system: current knowledge and future insights”, page 8, lines 266-279).

I'm totally agree with the conclusion (one-fit-for-all position) and that clearly need to be highlighted. Personalized prescription need to be the standard and effects on immune system need to be included. You could maybe suggest a work up and a decisional algorithm for dialysis prescription, that could be great.

We included in the revised version an extra table (Table 2 at page 9), detailing the decisional criteria currently used in our Center for dialysis prescription.

Minor comment.

Table 1: put it on a landscape special page to upgrade the readability.

Thanks, this has been done (page 7).

Round 2

Reviewer 1 Report

This revised version of the manuscript has taken into account redviewers'comments and seems suitable for publication.

Reviewer 2 Report

Ok with the new version

Thanks